# SOC: Semantic-Assisted Object Cluster for Referring Video Object Segmentation

**Zhuoyan Luo**[1*], **Yicheng Xiao**[1*], **Yong Liu**[1*],
**Shuyan Li**[3], **Yitong Wang**[2], **Yansong Tang**[1], **Xiu Li**[1†], **Yujiu Yang**[1†]

[1]Tsinghua Shenzhen International Graduate School, Tsinghua University
[2]ByteDance Inc.
[3]Engineering Department, University of Cambridge
{luozy23, xiaoyc23, liu-yong20}@mails.tsinghua.edu.cn
sl2141@cam.ac.uk    wangjingshen@bytedance.com
{tang.yansong, li.xiu, yang.yujiu}@sz.tsinghua.edu.cn

## Abstract

This paper studies referring video object segmentation (RVOS) by boosting video-level visual-linguistic alignment. Recent approaches model the RVOS task as a sequence prediction problem and perform multi-modal interaction as well as segmentation for each frame separately. However, the lack of a global view of video content leads to difficulties in effectively utilizing inter-frame relationships and understanding textual descriptions of object temporal variations. To address this issue, we propose **S**emantic-assisted **O**bject **C**luster (SOC), which aggregates video content and textual guidance for unified temporal modeling and cross-modal alignment. By associating a group of frame-level object embeddings with language tokens, SOC facilitates joint space learning across modalities and time steps. Moreover, we present multi-modal contrastive supervision to help construct well-aligned joint space at the video level. We conduct extensive experiments on popular RVOS benchmarks, and our method outperforms state-of-the-art competitors on all benchmarks by a remarkable margin. Besides, the emphasis on temporal coherence enhances the segmentation stability and adaptability of our method in processing text expressions with temporal variations. Code is available at https://github.com/RobertLuo1/NeurIPS2023_SOC.

## 1 Introduction

Referring Video Object Segmentation (RVOS) [1, 42] aims to segment the target object referred by the given text description in a video. Unlike conventional single-modal segmentation tasks according to pre-defined categories [31, 32] or visual guidance [24–26, 10], referring segmentation requires comprehensive understanding of the content across different modalities to identify and segment the target object accurately. Compared to referring image segmentation, RVOS is even more challenging since the algorithms must also model the temporal relationships of different objects and locations. This emerging topic has attracted great attention and has many potential applications, such as video editing and human-robot interaction.

Due to the varieties of video content as well as the unrestricted language expression, the critical problem of RVOS lies in how to perform pixel-level alignment between different modalities and time steps. To accomplish this challenging task, previous methods have tried various alignment workflows.

---

[*]Equal contribution. † Corresponding author.

37th Conference on Neural Information Processing Systems (NeurIPS 2023).

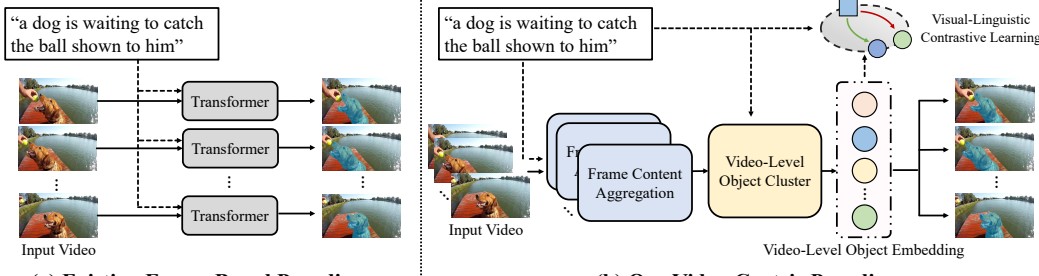

**(a) Existing Frame-Based Paradigm**        **(b) Our Video-Centric Paradigm**

Figure 1: Illustration of different paradigms. Frame-based methods perform cross-modal interaction and segmentation for each frame individually. In contrast, our method unifies temporal modeling and cross-modal alignment to achieve video-level understanding.

Early approaches [5, 9, 14, 20, 28, 44, 48] take the bottom-up or top-down paradigms to segment each frame separately, while recent works [1, 42] propose to unify cross-modal interaction with pixel-level understanding into transformer structure. Although the above-mentioned approaches have facilitated the alignment between different modalities and achieved excellent performance, they model the RVOS task as a sequence prediction problem and pay little attention to the temporal relationships between different frames. Specifically, they perform cross-modal interaction and segmentation for each frame individually, as illustrated in Fig. 1 (a). The exploitation of temporal guidance relies on the spatial-temporal backbone and manually designed hard assignment strategies [1, 42]. Such paradigms convert the referring video segmentation into stacks of referring image segmentation. While it may be acceptable for such conversion to handle the descriptions of static properties such as the appearance and color of the objects, this approach may lose the perception of target objects for language descriptions expressing temporal variations of objects due to the lack of video-level multi-modal understanding.

To alleviate the above problems and align video with text effectively, we propose Semantic-assisted Object Cluster (SOC) to perform object aggregation and promote visual-linguistic alignment at the video level, as depicted in Fig. 1 (b). Specifically, we design a Semantic Integration Module (SIM) to efficiently aggregate intra-frame and inter-frame information. With a global view of the video content, SIM can facilitate the understanding of temporal variations as well as alignment across different modalities and granularity. Furthermore, we introduce visual-linguistic contrastive learning to provide semantic supervision and guide the establishment of video-level multi-modal joint space. In addition to the remarkable improvements in generic scenarios, these efforts also allow our method to effectively handle text descriptions expressing temporal variations.

We conduct experiments on popular RVOS benchmarks, *i.e.*, Ref-YouTube-VOS [36], Ref-DAVIS [16], A2D-Sentences and JHMDB-Sentences [8], to validate the effectiveness of our method. Results show that SOC notably outperforms existing methods for all benchmarks with faster inference speed. In addition, we provide detailed ablations and analysis on components of our method.

Overall, our contributions are summarized as follows:

- We present a framework called SOC for RVOS to unify temporal modeling and cross-modal alignment. In SOC, a Semantic Integration Module (SIM) is designed to efficiently aggregate inter and intra-frame information, which achieves video-level multi-modal understanding.

- We introduce a visual-linguistic contrastive loss to apply semantic supervision on video-level object representations, resulting in well-aligned multi-modal joint space.

- Without bells and whistles, our method outperforms existing state-of-the-art method Refer-Former [42] by a remarkable margin, *e.g.*, +3.0% $\mathcal{J}\&\mathcal{F}$ on Ref-YouTube-VOS and +3.8% $\mathcal{J}\&\mathcal{F}$ on Ref-DAVIS under fair comparison. Besides, our method runs at 32.3 FPS on single 3090 GPU, which is significantly faster than the 21.4 FPS of ReferFormer.

## 2   Related Work

**Referring Image Segmentation**   Referring Image Segmentation (RIS) aims to localize the corresponding object referred by a text description within a static image. It is first introduced by Hu *et*

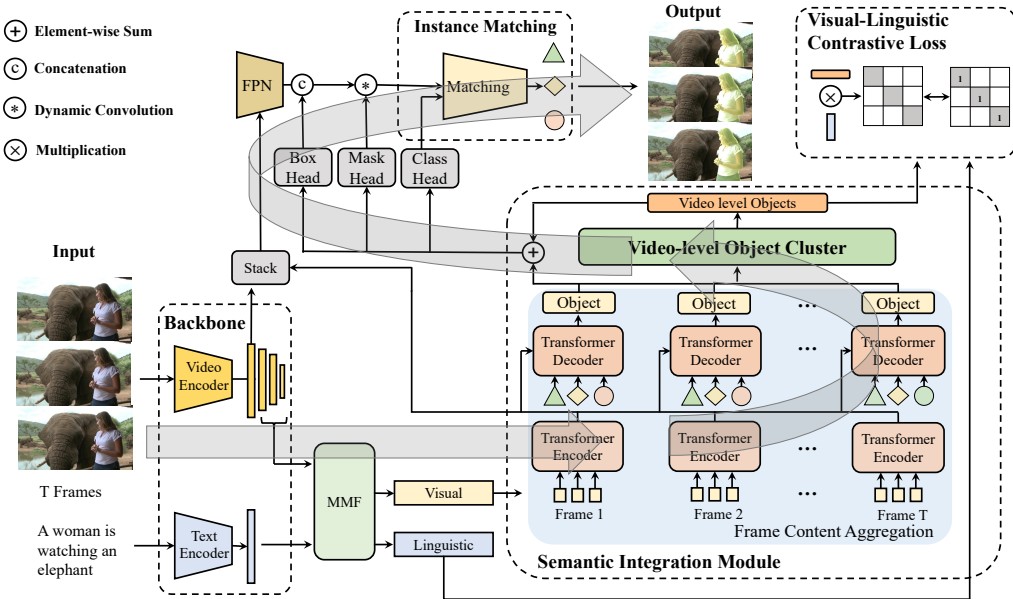

Figure 2: Overview of SOC. The model takes a video clip with corresponding language descriptions as input. After the encoding process, the multi-modal fusion (MMF) module performs bidirectional fusion to build the intrinsic feature relations. Then we design a Semantic Integration Module to efficiently aggregate intra-frame and inter-frame information. Meanwhile, we introduce a visual-linguistic contrastive loss to benefit the establishment of video-level multi-modal space. Finally, the prediction heads decode the condensed embeddings and output segmentation masks. The transparent arrows illustrates the pipeline of our model and label the input and output.

*al.* [12], who develops a simple framework that utilizes Fully Convolution Network (FCN) [37] to generate segmentation masks from concatenated visual and linguistic features. To deeply explore the intrinsic correlations among different modal features, several studies [3, 13, 41, 46] design various attention modules for modality interaction. Additionally, VLT [4] proposes a transformer-based architecture for the RIS task, which has gained more popularity than FCN-based approaches. LAVT [45] incorporates early alignment of visual and linguistic features at the intermediate layers of encoders. PolyFormer [21] further uses transformer to generate polygon vertices as the prior information to refine segmentation masks, leading to better results.

**Referring Video Object Segmentation**   Compared to RIS, RVOS is more challenging since both the action and appearance of the referred object are required to be segmented in a dynamic video. Gavri-lyuk *et al.* [8] first proposes the Referring Video Object Segmentation (RVOS) task. URVOS [36] introduces a large-scale RVOS benchmark and a unified framework that leverages attention mechanisms and mask propagation to increase the task's complexity and scope. ACAN [40] designs an asymmetric cross-guided attention network to establish complex visual-linguistic relationships. To improve positional relation representations in the text, PRPE [35] explores a positional encoding mechanism based on the polar coordinate system. In addition, most previous approaches [19, 22, 39, 47, 48, 30] rely on complicated pipelines. To simplify the workflow, MTTR [1] and ReferFormer [42] adopt query-based end-to-end frameworks for decoding objects from multi-modal features, achieving excellent performance. However, during the decoding phase, previous methods only concentrate on intra-frame object information, disregarding the valuable temporal context of objects across frames. To address this issue, we propose to associate object temporal context with language tokens and achieve video-level multi-modal understanding.

## 3   Method

Given $T$ frames of video clip $\mathcal{I} = \{I_t\}_{t=1}^{T}$, where $I_t \in \mathbb{R}^{3 \times H_0 \times W_0}$ and a referring text expression $\mathcal{E} = \{e_i\}_{i=1}^{L}$, where $e_i$ denotes the i-th word in the text. Our goal is to generate a series of binary

segmentation masks $\mathcal{S} = \{s_t\}_{t=1}^T$, $s_t \in \mathbb{R}^{1 \times H_0 \times W_0}$ of the referred object. To this end, we propose a video-centric framework called Semantic-assisted Object Cluster (SOC). We will elaborate on it in the following sections.

## 3.1 Visual and Linguistic Encoding

**Visual Encoder**    Taking a video clip $\mathcal{I}$ as input, we utilize a spatial-temporal backbone such as Video Swin Transformer [27] to extract hierarchical vision features. Consequently, the video clip is encoded into a set of feature maps $\mathcal{F}_i^v \in \mathrm{R}^{C_i \times H_i \times W_i}$, $i \in \{1, 2, 3, 4\}$. Here $H_i$ and $W_i$ denote the height and width of each scale feature map, respectively. $C$ denotes the channel dimension.

**Language Encoder**    Simultaneously, a transformer-based [38] language encoder encodes the given textual expression $\mathcal{E}$ to a word-level embedding $\mathcal{F}^w \in \mathbb{R}^{L \times C_t}$ and a sentence-level embedding $\mathcal{F}^s \in \mathbb{R}^{1 \times C_t}$. The word embedding $\mathcal{F}^w$ contains fine-grained description information. On the other hand, the sentence embedding $\mathcal{F}^s$ expresses the general characteristics of the referred target object.

## 3.2 Two Stream Multi-Modal Fusion

Having the separate visual and linguistic embedding encoded from the video clip and text expression, we design a Multi-Modal Fusion module called MMF to perform preliminary cross-modal alignment. As shown in Fig. 3, MMF is a two-stream structure. The language-to-vision (L2V) stream aims to highlight the corresponding regions of the referred object in each frame and mitigate the effect of background noise. It leverages linguistic information as guidance and addresses the potential similar visual areas. Meanwhile, a vision-to-language (V2L) stream is designed to update the textual embedding with image content, which helps to relieve the potential ambiguity of unconstrained descriptions. Specifically, we measure the relevance of all visual areas to the text query and assign weights to the useful information extracted from the visual features so as to reorganize the text embedding. The above L2V and V2L fusion process are based on the multi-head cross-attention mechanism, which can be formulated as:

$$\mathrm{MHA}\left(\mathcal{X}, \mathcal{Y}\right) = \mathrm{Concat}\left(head_1\left(\mathcal{X}, \mathcal{Y}\right), \ldots, head_h\left(\mathcal{X}, \mathcal{Y}\right)\right) W,$$

$$head_j\left(\mathcal{X}, \mathcal{Y}\right) = \mathrm{softmax}\left(\frac{\left(\mathcal{X}W_j^Q\right)^T \mathcal{Y}W_j^K}{\sqrt{C}}\right) \mathcal{Y}W_j^V, \tag{1}$$

where $\mathrm{MHA}\left(\cdot\right)$ stands for multi-head attention. $W, W_j^Q, W_j^K, W_j^V$ are learnable weights used to map the input to the attention space.

Since visual features of different scales contain diverse content information, MMF is designed to produce a series of coarse-to-fine visual and textual feature maps $\{\mathcal{F}_i^{vf}\}$ and $\{\mathcal{F}_i^{ef}\}$, $i \in \{2, 3, 4\}$. Specifically, we leverage shared parameters to perform multi-head cross-attention operations on $\{\mathcal{F}_i^v\}$, $i \in \{2, 3, 4\}$ and $\mathcal{F}^w$. Take $\mathcal{F}_2^v$ as an example. Firstly, we utilize $1 \times 1$ convolution and fully connected layers to transform the visual and linguistic embeddings into joint space, respectively. Then the bidirectional multi-modal fusion is applied to align information from different modalities as well as enhance the feature representation. The fusion process is:

$$\mathcal{F}_2^{vf} = \mathrm{MHA}\left(\mathcal{F}_2^v, \mathcal{F}^w\right) \cdot \mathcal{F}_2^v, \tag{2}$$

$$\mathcal{F}_2^{ef} = \mathrm{MHA}\left(\mathcal{F}^w, \mathcal{F}_2^v\right) \cdot \mathcal{F}^w, \tag{3}$$

where $\mathcal{F}_2^v \in \mathbb{R}^{TH_2W_2 \times D}$ and $\mathcal{F}^w \in \mathbb{R}^{L \times D}$ are embeddings projected by convolution and fully connected layers. Similar to Eq. (2) and Eq. (3), the coarse-to-fine aligned visual features $\{\mathcal{F}_i^{vf}\}$ and textual embeddings $\{\mathcal{F}_i^{ef}\}$, $i \in \{2, 3, 4\}$ are produced.

## 3.3 Semantic Integration Module

After aligning cross-modal information and activating the potential target region in MMF, we design a Semantic Integration Module (SIM) to incorporate visual content and generate compact target representations. Specifically, we first leverage frame-level content aggregation to locate objects separately for each frame. Then we associate inter-frame dependency and model the temporal relationship of objects via video-level object cluster.

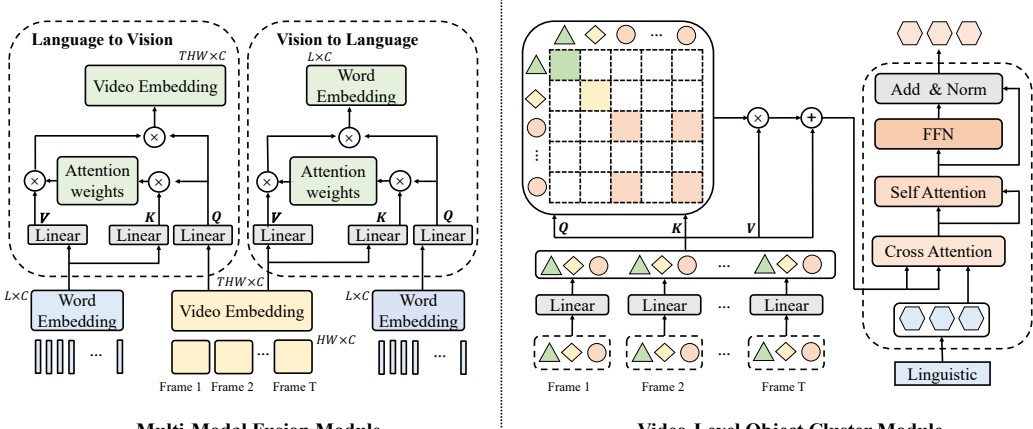

Figure 3: The structure of proposed Multi-Modal Fusion module (MMF) and Video-level Object Cluster module (VOC).

**Frame-Level Content Aggregation** Having the activated visual features $\{\mathcal{F}_i^{vf}\}$ from MMF, we utilize a transformer-based structure to locate objects in each frame. Firstly, $K$ stacks of deformable transformer encoder layer [49] are leveraged to capture intra-frame relationships and further excavate multi-modal interactions inside $\{\mathcal{F}_i^{vf}\}$. This process can be formulated as:

$$\mathcal{F'}_i^{vf} = \left\{\text{DecformEnc}_k\left(f_t^{vf}\right)\right\}_{t=1}^T, \tag{4}$$

where $f_t^{vf}$ denotes the activated visual features of the t-th frame. Then, a set of learnable object queries [2, 49, 34] is introduced to aggregate image content and highlight potential target objects. Following [42], these object queries fully interact with $\mathcal{F'}_i^{vf}$ in deformable transformer decoder through cross-attention mechanism. After extracting different object representations, these object queries are turned into instance embeddings $\mathcal{O}^f \in \mathbb{R}^{T \times N_q \times D}$. Note that we set up $N_q$ object queries to represent instances of each frame in a video clip so there are $N_q T$ output object queries in total.

**Video-Level Object Cluster** Instances typically vary in pose and location between frames and even being obscured. Previous methods [1, 5, 6, 42] only model instances separately for each frame, disregarding the temporal relationship and continuous motion of instances. Although such an approach can cover simple scenarios such as descriptions of target appearance, the lack of inter-frame interaction makes existing methods inefficient for the description of temporal relationships. To address the aforementioned weakness, inspired by [7, 11, 18], we design a video-level object cluster to capture the temporal information of instances between frames.

As shown in Fig. 3, after instances embedding $\mathcal{O}^f$ are formulated by the frame-level content aggregation, we flatten it into $\mathcal{O}^{f'} \in \mathbb{R}^{TN_q \times D}$ and employ self-attention mechanism along the temporal axis to introduce inter-frame interaction. Furthermore, we find that only introducing temporal self-attention is inferior. Simply sharing temporal object context may create redundancy due to the similarity of the representation referring to the same object in different frames, which potentially affects the model's precise understanding of objects in the video clip. To this end, we employ an object grouping decoder to perform object clustering and group the same object into video-level compact representations across frames. Specifically, we introduce $N_v$ video-level object queries $\mathcal{O}^v \in \mathbb{R}^{N_v \times D}$ and they are initialized by linguistic sentence-level feature $\mathcal{F}^s$, which helps to promote the understanding of object descriptions and build the visual-linguistic joint space. By performing interaction between linguistic-aware video queries and condensed instance information of each frame, the object grouping decoder can effectively capture temporal object contexts and group the referred object queries across frames to output the clustered video-level object queries $\mathcal{O}^v$. While temporal connections can provide a lot of valuable information, the segmentation of each frame is dependent on the specific image content. Thus, we incorporate the high-level video instance information into each frame to integrate the advantages of both. In detail, as illustrated in Fig. 2, we repeat the video-level object queries $T$ times $\mathcal{O}'^v \in \mathbb{R}^{T \times N_v \times D}$ and enhance the representation of frame-level object queries

by element-wise sum: $\mathcal{O}^f = \mathcal{O}^f + \mathcal{O}'^v$. In this way, the semantic of frame-level object queries is greatly enriched by the supplement of video-level object information.

## 3.4 Visual-Linguistic Contrastive Learning

Although the aggregated video-level representation can better describe object states, the simple exploitation of textual priors may lead to non-target potential responses for video-level embeddings [29]. Meanwhile, taking unrestricted expressions as aggregation guidance may introduce undesirable inductive bias. To address the issues, we present a visual-linguistic contrastive loss that explicitly focuses on the alignment between video-level features and textual information. The loss accompanies two purposes: (1) Bridge the semantic gap between textual expressions and the corresponding object queries in the multi-modal joint embedding space. (2) Mitigate the bias caused by unrestricted textual features and emphasize the referred object representations. As shown in Fig. 2, we transform the last stage of fused textual features $\mathcal{F}_4^{ef}$ from the vision-to-language stream (see Eq. (3)) to generate the textual guidance embedding $\mathcal{F}_{gud}$:

$$\mathcal{F}_{gud} = \mathrm{AveragePooling}\left(\mathcal{F}_4^{ef}\right), \mathcal{F}_{gud} \in \mathbb{R}^D. \tag{5}$$

Then, we measure the similarity between the textual guidance embedding and video-level object queries by scaled dot product similarity: $\hat{y}_{sim} = \frac{\mathcal{O}^v \mathcal{F}_{gud}^\top}{\sqrt{D}}$. To suppress undesired region response and highlight target object, we take softmax operation along the object query axis. Finally, we compute the contrastive loss $\mathcal{L}_{con}$ by matrix multiplication of $\hat{y}_{sim}$ and $y_\tau$, where $y_\tau$ is annotated to 1 for the best predicted trajectory and 0 for others [43]:

$$\mathcal{L}_{con} = -\mathrm{LogSoftmax}\left(\hat{y}_{sim}\right) \cdot y_\tau. \tag{6}$$

## 3.5 Instance Segmentation and Loss

**Prediction Heads**  As depicted in Fig. 2, there are three lightweight heads built on top of the semantic integration module. The classification head is a concatenation of three fully connected layers. It directly predicts class probability $\hat{p} \in \mathbb{R}^{T \times N_q \times (K+1)}$ for frame-level object queries, where $K$ is the number of classes. Note that if $K = 0$, the role of the classification head is to judge whether the object is referred by the text description. The box head comprises three sequential linear layers, which is designed to transform the object queries to normalized bounding box information $\hat{b} \in \mathbb{R}^{T \times N_q \times 4}$, *i.e.*, center coordinates, width, and height. Similar to [42], we adopt dynamic convolution to output segmentation masks for each frame. Specifically, the mask head produces weights of $N_q$ dynamic kernels $\Omega = \{\omega_n\}_{n=1}^{N_q}$. Meanwhile, the encoded multi-scale cross-modal features $\{\mathcal{F}'^{vf}_i\}$ are stacked with the $4\times$ features from the visual backbone to form hierarchical decoding features. An FPN structure takes the hierarchical features as input and outputs high-resolution semantic-aware features $\mathcal{F}_{seg} \in \mathbb{R}^{T \times \frac{H_0}{4} \times \frac{W_0}{4} \times D}$. Finally, the dynamic kernels process the features $\mathcal{F}_{seg}$ based on the position information output from the box head and obtain the segmentation mask $\hat{m} \in \mathbb{R}^{T \times N_q \times \frac{H_0}{4} \times \frac{W_0}{4}}$, which can be formulated as:

$$\hat{m}_n = \left\{\hat{f}_n^{seg} \circledast \omega_i\right\}_{n=1}^{N_q}. \tag{7}$$

**Instance Matching**  As described above, the prediction heads output the prediction trajectories of $N_q$ objects, denoted as $\hat{y} = \{\hat{y}_n\}$ and the n-th object trajectory prediction is:

$$\hat{y}_n = \{\hat{p}_n^t, \hat{b}_n^t, \hat{m}_n^t\}_t^T. \tag{8}$$

There is only one referred object in a video clip corresponding to the text descriptions. Therefore, we denote the ground truth object sequence as $y = \{p^t, b^t, m^t\}_t^T$ and search the best prediction trajectory $\hat{y}_\sigma$ via Hungarian algorithm [17].

**Total Loss**  We supervise the trajectory prediction $\hat{y}_\sigma$ by four types of losses: (1) $\mathcal{L}_{mask}(y, \hat{y}_\sigma)$, the mask loss is a combination of Dice loss and binary focal loss, which is computed across frames. (2) $\mathcal{L}_{box}(y, \hat{y}_\sigma)$: the box loss aggregates the L1 loss and GIoU loss per-frame. (3) $\mathcal{L}_{cls}(y, \hat{y}_\sigma)$: the class

| Method | Backbone | Ref-YouTube-VOS | | | Ref-DAVIS17 | | |
|--------|----------|-----------------|--|--|-------------|--|--|
| | | $\mathcal{J}\&\mathcal{F}$ | $\mathcal{J}$ | $\mathcal{F}$ | $\mathcal{J}\&\mathcal{F}$ | $\mathcal{J}$ | $\mathcal{F}$ |
| URVOS [36] | ResNet-50 | 47.2 | 45.3 | 49.2 | 51.5 | 47.3 | 56.0 |
| LBDT-4 [6] | ResNet-50 | 49.4 | 48.2 | 50.6 | - | - | - |
| MTTR [1] | Video-Swin-T | 55.3 | 54.0 | 56.6 | - | - | - |
| ReferFormer [42] | Video-Swin-T | 56.0 | 54.8 | 57.3 | - | - | - |
| SOC (Ours) | Video-Swin-T | **59.2** | **57.8** | **60.5** | **59.0** | **55.4** | **62.6** |
| *With Image Pretrain* | | | | | | | |
| ReferFormer [42] | Video-Swin-T | 59.4 | 58.0 | 60.9 | 59.7 | 56.6 | 62.8 |
| ReferFormer [42] | Video-Swin-B | 62.9 | 61.3 | 64.6 | 61.1 | 58.1 | 64.1 |
| VLT [5] | Video-Swin-B | 63.8 | 61.9 | 65.6 | 61.6 | 58.9 | 64.3 |
| SOC (Ours) | Video-Swin-T | **62.4** | **61.1** | **63.7** | **63.5** | **60.2** | **66.7** |
| SOC (Ours) | Video-Swin-B | **66.0** | **64.1** | **67.9** | **64.2** | **61.0** | **67.4** |
| *Joint Train* | | | | | | | |
| ReferFormer | Video-Swin-T | 62.6 | 59.9 | 63.3 | - | - | - |
| ReferFormer | Video-Swin-B | 64.9 | 62.8 | 67.0 | - | - | - |
| SOC (Ours) | Video-Swin-T | **65.0** | **63.3** | **66.7** | **64.2** | **60.9** | **67.5** |
| SOC (Ours) | Video-Swin-B | **67.3** | **65.3** | **69.3** | **65.8** | **62.5** | **69.1** |

Table 1: Comparison with the state-of-the-art methods on Ref-YouTube-VOS and Ref-DAVIS17 datasets. *With Image Pretrain* denotes the models are first pretrained on RefCOCO [15], Ref-COCO+ [15], and RefCOCOg [33] datasets. *Joint Train* indicates the models are trained with the combination of image datasets and video datasets.

loss is focal loss and supervises the predicted object category. (4) $\mathcal{L}_{con}(y_\tau, \hat{y}_{sim})$: visual-linguistic contrastive loss. The total loss can be formulated as:

$$\mathcal{L}_{total} = \lambda_{mask}\mathcal{L}_{mask} + \lambda_{box}\mathcal{L}_{box} + \lambda_{cls}\mathcal{L}_{cls} + \lambda_{con}\mathcal{L}_{con}, \tag{9}$$

where $\lambda$ is the scale factor to balance each loss.

## 4 Experiment

### 4.1 Datasets and Metrics

**Datasets.** We evaluate our model on four prevalent RVOS benchmarks: Ref-YouTube-VOS [36], Ref-DAVIS17 [16], A2D-Sentences, and JHMDB-Sentences [8]. For detailed descriptions of the datasets please see the supplementary material.

**Metrics.** Following [1, 42], we measure the effectiveness of our model by criteria of Precision@K, Overall IoU, MeanIoU and MAP over 0.50:0.05:0.95 for A2D-Sentences and JHMDB-Sentences. Meanwhile, we adopt standard evaluation metrics: region similarity($\mathcal{J}$), contour accuracy ($\mathcal{F}$) and their average value ($\mathcal{J}\&\mathcal{F}$) on Ref-YouTube-VOS and Ref-DAVIS17.

### 4.2 Implementation Details

We take the pretrained Video Swin Transformer [27] and RoBERTa [23] as our encoder in default. Both the frame aggregation and object cluster parts of SIM consist of three encoder and decoder layers. The number of frame-level queries $\mathcal{O}^f$ and video-level queries $\mathcal{O}^v$ are set as 20 in default. We feed the model windows of $w = 8$ frames during training. The models are trained with eight 32GB V100 GPUs in default. The coefficients for losses are set as $\lambda_{cls} = 2$, $\lambda_{L1} = 2$, $\lambda_{giou} = 2$, $\lambda_{dice} = 2$, $\lambda_{focal} = 5$, $\lambda_{con} = 1$. Due to space limitations, please see the supplementary materials for more training details.

### 4.3 Main Results

**Ref-YouTube-VOS & Ref-DAVIS17.** We compare our method to previous models on Ref-YouTube-VOS and Ref-DAVIS17 in Table 1. With video-level multi-modal understanding, our SOC achieves new state-of-the-art performance among different training settings: train from scratch, with image pretrain, and joint train. Without bells and whistles, our approach outperforms existing SOTA by about 3% $\mathcal{J}\&\mathcal{F}$ under fair comparison. On Ref-DAVIS17, we directly report the results using the model trained on Ref-YouTube-VOS without finetune.

| Method | Backbone | Precision | | | | | IoU | | mAP |
|---|---|---|---|---|---|---|---|---|---|
| | | P@0.5 | P@0.6 | P@0.7 | P@0.8 | P@0.9 | Overall | Mean | |
| Hu *et al.*. [12] | VGG-16 | 34.8 | 23.6 | 13.3 | 3.3 | 0.1 | 47.4 | 35.0 | 13.2 |
| Gavrilyuk *et al.* [8] | I3D | 47.5 | 34.7 | 21.1 | 8.0 | 0.2 | 53.6 | 42.1 | 19.8 |
| CMSA + CFSA [47] | ResNet-101 | 48.7 | 43.1 | 35.8 | 23.1 | 5.2 | 61.8 | 43.2 | - |
| ACAN [40] | I3D | 55.7 | 45.9 | 31.9 | 16.0 | 2.0 | 60.1 | 49.0 | 27.4 |
| CMPC-V [22] | I3D | 65.5 | 59.2 | 50.6 | 34.2 | 9.8 | 65.3 | 57.3 | 40.4 |
| ClawCraneNet [19] | ResNet-50/101 | 70.4 | 67.7 | 61.7 | 48.9 | 17.1 | 63.1 | 59.9 | - |
| MTTR [1] | Video-Swin-T | 75.4 | 71.2 | 63.8 | 48.5 | 16.9 | 72.0 | 64.0 | 46.1 |
| ReferFormer [42] | Video-Swin-T | 76.0 | 72.2 | 65.4 | 49.8 | 17.9 | 72.3 | 64.1 | 48.6 |
| SOC (Ours) | Video-Swin-T | **79.0** | **75.6** | **68.7** | **53.5** | **19.5** | **74.7** | **66.9** | **50.4** |
| *With Image Pretrain* | | | | | | | | | |
| ReferFormer [42] | Video-Swin-T | 82.8 | 79.2 | 72.3 | 55.3 | 19.3 | 77.6 | 69.6 | 52.8 |
| ReferFormer [42] | Video-Swin-B | 83.1 | 80.4 | 74.1 | 57.9 | 21.2 | 78.6 | 70.3 | 55.0 |
| SOC (Ours) | Video-Swin-T | **83.1** | **80.6** | **73.9** | **57.7** | **21.8** | **78.3** | **70.6** | **54.8** |
| SOC (Ours) | Video-Swin-B | **85.1** | **82.7** | **76.5** | **60.7** | **25.2** | **80.7** | **72.5** | **57.3** |

Table 2: Comparison with the state-of-the-art methods on A2D-Sentences.

| VOC | VL | $\mathcal{J}\&\mathcal{F}$ | $\mathcal{J}$ | $\mathcal{F}$ |
|---|---|---|---|---|
| | | 55.6 | 54.3 | 56.9 |
| ✓ | | 58.0 | 56.5 | 59.6 |
| | ✓ | 57.1 | 55.8 | 58.4 |
| ✓ | ✓ | **59.2** | **57.8** | **60.5** |

Table 3: Effectiveness of our proposed modules.

| Fusion Strategy | $\mathcal{J}\&\mathcal{F}$ | $\mathcal{J}$ | $\mathcal{F}$ |
|---|---|---|---|
| No Fusion | 38.3 | 36.1 | 40.6 |
| V2L | 39.8 | 37.6 | 42.0 |
| L2V | 58.7 | 57.3 | 60.0 |
| Both | **59.2** | **57.8** | **60.5** |

Table 4: Impact of different fusion strategies.

| VOC Method | $\mathcal{J}\&\mathcal{F}$ | $\mathcal{J}$ | $\mathcal{F}$ |
|---|---|---|---|
| K-Means | 33.8 | 32.9 | 34.7 |
| 1D-Conv | 55.1 | 53.8 | 56.4 |
| Average | 57.5 | 56.1 | 58.9 |
| Ours | **59.2** | **57.8** | **60.5** |

Table 5: Impact of different cluster methods.

| Query Num. | $\mathcal{J}\&\mathcal{F}$ | $\mathcal{J}$ | $\mathcal{F}$ |
|---|---|---|---|
| 10 | 58.0 | 56.6 | 59.5 |
| 15 | 58.1 | 56.7 | 59.6 |
| 20 | **59.2** | **57.8** | **60.5** |
| 25 | 58.5 | 57.2 | 59.8 |

Table 6: Results of different object query numbers.

| Frame Num. | $\mathcal{J}\&\mathcal{F}$ | $\mathcal{J}$ | $\mathcal{F}$ |
|---|---|---|---|
| 3 | 57.9 | 56.5 | 59.3 |
| 5 | 58.4 | 57.1 | 59.7 |
| 8 | **59.2** | **57.8** | **60.5** |
| 10 | 59.1 | 57.8 | 60.3 |

Table 7: Results of different training frame numbers.

| VOC Structure | $\mathcal{J}\&\mathcal{F}$ | $\mathcal{J}$ | $\mathcal{F}$ |
|---|---|---|---|
| None | 57.1 | 55.8 | 58.4 |
| only encoder | 58.3 | 56.9 | 59.8 |
| only decoder | 58.1 | 56.7 | 59.5 |
| Both | **59.2** | **57.8** | **60.5** |

Table 8: The structure design of the VOC module.

**A2D-Sentences & JHMDB-Sentences.** As shown in Table 2, compared with the existing SOTA method ReferFormer [42], our SOC achieves 50.4% mAP and 66.9% mIoU with the model trained from scratch, which gains a clear improvement of 1.8% mAP and 2.8% mIoU respectively. Furthermore, with image data pretraining, our method achieves new state-of-the-art on all metrics, *e.g.*, +4.0% P@0.9, +2.2% mIoU, and +2.3% mAP. Due to the similarity of the datasets and space limitations, please see the supplementary material for the comparison on JHMDB-Sentences [8]. Our SOC also surpasses all existing methods on it.

## 4.4 Ablation Studies

In this section, we conduct analysis experiments on the Ref-YouTube-VOS [36] benchmark using Video-Swin-Tiny as the visual backbone and train the model from scratch.

**Component Analysis** We experiment with the effectiveness of the proposed Video-level Object Cluster (VOC) and Visual-linguistic contrastive Learning (VL). As shown in Table 3, both of them bring performance improvement and their combination works better (+ 3.6% $\mathcal{J}\&\mathcal{F}$), which shows the significance of video-level multi-modal understanding.

**Fusion Strategy** We investigate the effects of various fusion strategies. As illustrated in Table 4, the absence of Language to Vision fusion (L2V) yields significant deterioration. This is because the frame-level content aggregation without textual guidance would introduce background noise. If only utilizing the L2V fusion, the performance drops 0.5% $\mathcal{J}\&\mathcal{F}$, demonstrating that visual content helps filter out irrelevant words from unconstrained descriptions.

**VOC Method** The superiority of the current VOC module derives from two aspects: i) Powerful representation learning ability of the transformer structure. Compared to other clustering methods

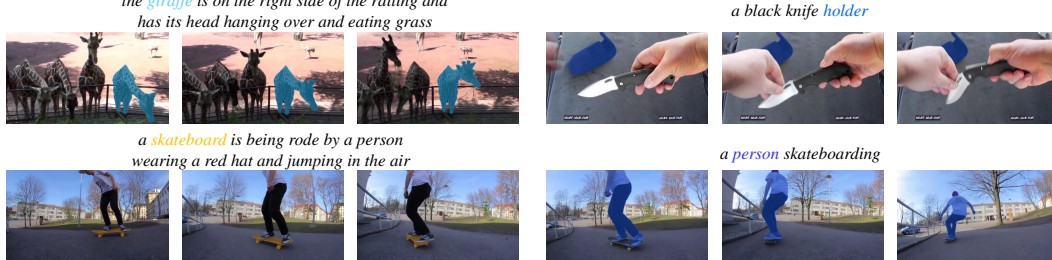

*the giraffe is on the right side of the railing and has its head hanging over and eating grass*

*a black knife holder*

*a skateboard is being rode by a person wearing a red hat and jumping in the air*

*a person skateboarding*

Figure 5: Segmentation results of our SOC. Best viewed in color.

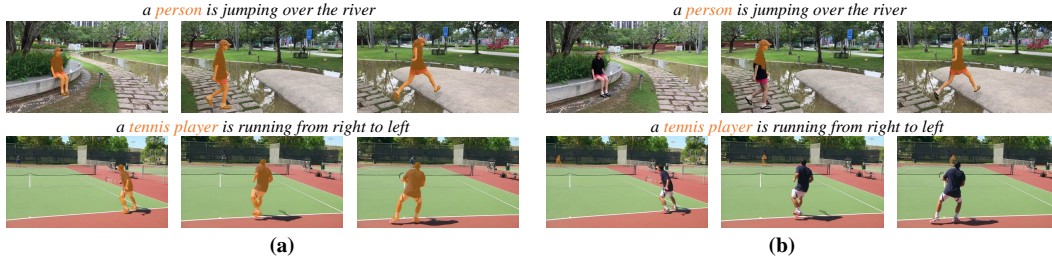

*a person is jumping over the river*

*a person is jumping over the river*

*a tennis player is running from right to left*

*a tennis player is running from right to left*

**(a)**              **(b)**

Figure 6: Visualization comparison using text expressions about temporal actions. (a) and (b) are segmentation results of our SOC and ReferFormer [42], respectively.

shown in Table 5, the transformer architecture has a better perception as well as selection of global information, which helps to associate object features across frames. ii) Sufficient inter-frame interactions. Although other methods can also yield video-level representations, they lack the perception of temporal variations and the alignment of inter-frame information. As a contrast, our VOC performs adequate inter-frame interaction while clustering.

**Object Query Number** Although only one referred object is involved in a video, focusing on more potential regions is helpful due to the various video content. Table 6 shows that an increased number of object queries allows the model to effectively group the relevant objects from the pool of candidate regions. The performance saturates when the query number is increased to a certain extent. We believe that excessive queries may cause misunderstandings when grouping objects across frames.

**Segmentation Stability** Benefiting from video-level multi-modal understanding, our method yields more robust segmentation results compared to existing approaches. To quantify the temporal segmentation stability, we calculate the IoU and $\mathcal{J}\&\mathcal{F}$ variance between each frame in a video and count the average value on Ref-YouTube-VOS [36] dataset. Results are shown in Fig. 4 (*the lower is better*). It can be seen that with video object cluster module and visual-linguistic contrastive loss building the video-level multi-modal aligned space, our SOC can understand the overall video content better and output

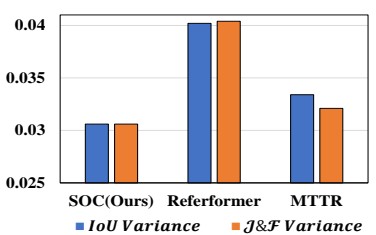

Figure 4: IoU and $\mathcal{J}\&\mathcal{F}$ variance.

temporally stable segmentation results. More qualitative analysis of temporal consistency is shown in the supplementary material.

**Frame Number** Table 7 shows results with different training clip frame numbers. As expected, widening temporal context leads to performance gains yet causes expensive computation cost. For a fair comparison with existing methods, we take the frame number of 8 by default.

**Variants of VOC Structure** Results in Table 8 illustrate the rationality of the structure of the VOC module. Without sufficient inter-frame interaction by the encoder or aggregation of target object information by the decoder, the segmentation performance will be degraded.

**Inference Speed** In addition to superior segmentation performance, our method also achieves real-time inference. Specifically, our SOC runs at 32.3 FPS on single 3090 GPU, which significantly exceeds the existing SOTA method ReferFormer [42] (21.4 FPS).

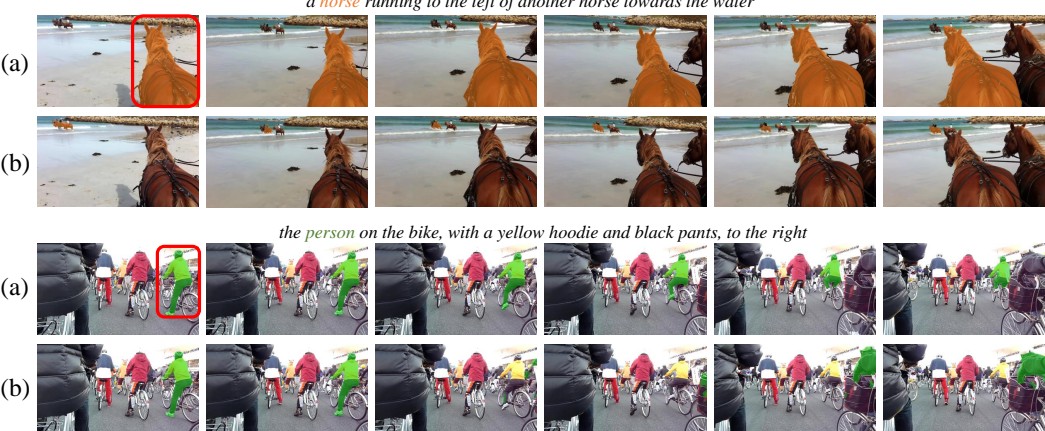

Figure 7: Visualization results of effectiveness on VOC. The object in the red box is the target. (a) displays the segmentation result of the completed model. While (b) shows the result of the model without VOC (Video Object Cluster in SIM).

## 4.5 Qualitative Results

Fig. 5 demonstrates the effectiveness of SOC for complex scenarios segmentation, *i.e.*, similar appearance, occlusion, and large variations. To further prove the *adaptability* of our SOC for understanding text expressions about temporal action and variations, we design corresponding text descriptions that depict changed states of the object in temporal. As shown in Fig. 6, the global view enables SOC to understand such text and segment the target object accurately. In contrast, ReferFormer [42] only recognizes the object in specific frames mentioned in the text descriptions and fails to comprehend the content. More comparisons can be seen in the supplementary material. Fig. 7 indicates that due to the temporal modeling and inter-frame interaction in VOC, the full model comprehends text descriptions of temporal variations and tracks the target in coherence. In contrast, without VOC, the model struggles to understand the temporal relationship and fails to segment the object.

## 5 Conclusion

In this paper, we propose a framework called SOC for RVOS to achieve video-level multi-modal understanding. By associating frame-level object embeddings with language tokens, we unify temporal modeling and cross-modal alignment into a simple architecture. Moreover, visual-linguistic contrastive learning is introduced to build the video-level multi-modal space. Extensive experiments show that our SOC remarkably outperforms existing state-of-the-art methods. Besides, video-level understanding also allows our SOC to handle text descriptions expressing temporal variations better.

**Limitations.** The proposed framework, SOC, has achieved video-level multi-modal understanding and excellent performance. However, there are some potential limitations, *e.g.*, the current architecture cannot handle infinitely long videos. We think that devising explicit designs for long videos with complex scenarios is an interesting future direction.

## Acknowledgements

This work was supported by the National Natural Science Foundation of China (Grant No. U1903213) and the Shenzhen Science and Technology Program (JSGG20220831110203007). This work was also supported by the National Key Research and Development Program of China under Grant 2020AAA0108302 and Shenzhen Stable Supporting Program (WDZC20200820200655001).

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
