# OpenReview forum: "SOC: Semantic-Assisted  Object Cluster for Referring Video Object Segmentation"
_NeurIPS.cc/2023/Conference — NeurIPS 2023 poster_

### Official Review · Reviewer_9DMq · 2023-07-04

**Soundness:** 3 good
**Presentation:** 3 good
**Contribution:** 3 good
**Rating:** 7
**Confidence:** 2

**Summary:**

This paper proposes a novel framework that can effectively exploit video-level visual-linguistic alignment to improve the performance of RVOS, filling a gap in existing methods. The extensive experiments on three datasets demonstrate  the effectiveness of SOC.

**Strengths:**

The proposed method outperforms existing state-of-the-art methods on all datasets.
The paper provides a clear and concise description of the SOC framework and its experimental results.

**Weaknesses:**

None

**Questions:**

How to handle objects that have been occluded for a short time (> training frames number)

**Limitations:**

The authors have addressed the limitations and potential negative societal impact of their work.

---

> ### Author Rebuttal · Authors · 2023-08-09
>
> We sincerely thank the Reviewer 9DMq for his/her comments and the appreciation of our work. Our response to the Reviewer 9DMq is as follows:
>
> *Questions:* Thanks for your question. Sorry we are not sure whether you are referring to the training stage or the inference. Therefore, we explain both of these two stage. During the training stage, if objects are occluded, the associated ground truth masks will also have no labeled area and would not impede the training process. For the inference stage, since we input the entire video and perform multi-modal understanding at the video level, the model is not affected by the number of training frames. During inference, occlusion scenes tend to severely affect the feature representations. But information gleaned from other frames where the occluded object is not obscured aids in localizing and segmenting target in specific frames. Thus, our method also performs well for occlusion scenarios by fusing intra-frame and inter-frame information. Segmentation visualization of the occluded objects by our method is also provided in the supplementary material. We wonder if the above response addresses your questions. If you still have any questions about our response, we would be pleased to discuss them with you.

---

### Official Review · Reviewer_d9RC · 2023-07-06

**Soundness:** 3 good
**Presentation:** 3 good
**Contribution:** 2 fair
**Rating:** 5
**Confidence:** 4

**Summary:**

This work studies the referring video object segmentation problem and proposes the semantic-assisted object cluster approach. It aggregates video content and textual guidance for unified temporal modeling and cross-modal alignment.

**Strengths:**

1) It performs object aggregation and promotes visual-linguistic alignment at the video level, which considers the temporal variations and alignment across different modalities of frames.

2) It introduces the visual-linguistic contrastive learning to provide semantic supervision.

3) Empirical studies on several benchmarks demonstrate the advantage of the proposed method.

**Weaknesses:**

1) The novelty seems a bit limited. The multi-modal fusion model just uses the common multi-head cross-attention to model the relations of language and visual information.

2) Is there other clustering methods for object grouping? Some discussions are requried.

3) Ablations studies on the hyper-parameters $\lambda$ are missing.

**Questions:**

What about considering the object-level temporal relations across frames compared to the video-level ones?

**Limitations:**

See the above.

---

> ### Author Rebuttal · Authors · 2023-08-08
>
> We sincerely thank the Reviewer d9RC for his/her comments and the appreciation of our work. To the concerns expressed in Weaknesses and Questions, our response is as follows:
>
> *Weakness-1:* Thanks for the question. MMF is just a component of our framework. It is not our contribution, and it is not mentioned in the contribution note in the introduction section of the paper. Our main contribution lies in the proposed framework, which unifies temporal modeling and cross-modal alignment to achieve video-level visual-linguistic understanding. By fusing intra-frame and inter-frame information, our approach achieves better performance and can handle descriptions of temporal variations. Regarding the MMF module specifically, it serves to activate textual referred objects in visual features and simultaneously update the textual embeddings by incorporating visual content.
>
> *Weakness-2:* Thanks for your valuable question. Table 7 in the manuscript shows the ablation of the structure of the current video-level object cluster (VOC) module.  Here we experiment with several intuitive methods for object grouping. Specifically, "Average" means global pooling of all frames-level queries at the corresponding positions. "1D-Conv" denotes that we use 1D convolution (kernel size is 5) to process the frame-level query and then pooling them. It can be seen that such methods are underperforming. We attribute this to the absence of sufficient inter-frame interaction. Specifically, although these methods can also yield video-level representations, they lack the perception of temporal variations and the alignment of inter-frame information. As a contrast, our VOC can take advantage of the transformer structure to accomplish inter-frame interactions while clustering. We will incorporate this part into the manuscript.
>
> | VOC Method | Average | K-Means | 1D-Conv | Ours |
> | --- | :---: | :---: | :---: | :---: |
> | $\mathcal{J} $&$ \mathcal{F}$| 57.5 | 33.8 | 55.1 | 59.2 |
>
>
>
> *Weakness-3:* Thanks for your question. Here we show the detailed ablation experiments for the loss weight $\lambda$.
>
> | $\lambda_{mask}$ | $\lambda_{box}$ | $\lambda_{cls}$ | $\lambda_{con}$ | $\mathcal{J} $&$ \mathcal{F}$ |
> | :---: | :---: | :---: | :---: | :---: |
> | 2| 5 | 2 | 1 | 59.2 |
> | 2| 5 | 1 | 1 | 58.3 |
> | 2| 5 | 1 | 2 | 58.5 |
> | 2| 2 | 1 | 1 | 58.6 |
> | 2| 2 | 2 | 2 | 57.9 |
> | 5| 2 | 2 | 1 | 59.1 |
>
> *Question:* Good question! We experiment with performing inter-frame interactions in different time scales (different temporal window size). Results show that larger temporal scale interaction is helpful and the video-level ones performs best. We believe the reason may be that the range of inter-frame interactions required for different textual descriptions is varied. In video-level interactions, the model can recognize and selectively extract information related to the target object, which leads to better performance as well as flexibility.
>
> | Temporal Window Size | $\mathcal{J} $&$ \mathcal{F}$ |
> | :---: | :---: |
> | 2 | 57.8 |
> | 4 | 58.2 |
> | 6 | 58.9 |
> | global | 59.2 |

---

> > ### Comment · Reviewer_d9RC · 2023-08-11
> >
> > Thanks for the rebuttal. Most of my concerns have been well clarified.

---

> > > ### Author Response · Authors · 2023-08-12
> > > **Official Comment by Authors**
> > >
> > > Dear Reviewer d9RC,
> > >
> > > Thanks for your response. We are glad that our rebuttal is able to address your concerns. We sincerely appreciate your time and effort in reviewing our paper. Your valuable comments are crucial in improving the quality of our work. If you still have any concerns, we would be pleased to discuss them further with you.
> > >
> > > Best regards,
> > >
> > > Paper 4195 Authors

---

### Official Review · Reviewer_ihRG · 2023-07-06

**Soundness:** 3 good
**Presentation:** 3 good
**Contribution:** 3 good
**Rating:** 6
**Confidence:** 4

**Summary:**

This research paper focuses on referring video object segmentation. It combines temporal frame features and language features using a cross attention design. Additionally, it extracts object features from each frame and applies contrastive learning with language representation. The paper presents various designs to enhance the referring performance. Results from experiments on multiple benchmarks demonstrate that this method achieves state-of-the-art performance on this task.



**Strengths:**

1. The RVOS framework demonstrates remarkable results on various benchmarks.
2. Through experimentation, the proposed fusion of features significantly enhances the baseline.
3. The concept of utilizing temporal and language features is effectively communicated.

**Weaknesses:**

1. The current method is quite complex, and it is not clear what the main contributions of the RVOS in figure are. It appears that most of the modules shown in Figure 1 are not mentioned in the manuscript. This makes it difficult to understand which proposed components are essential for the given task.

2. The Multi-Modal Fusion (MMF) and Semantic Integration Module (SIM) are presented in Section 3.2 and 3.3. However, there are no experimental tests conducted to validate these modules. The authors also claim that SIM contributes to the task in the introduction.

3. The results of visualizing the impact of the proposed modules are missing. It is important to provide results that demonstrate how each module enhances the referring performance.

**Questions:**

1. Figure 6 shows a video with only one person. Why does the compared method fail in these scenes? Are there any complex scenes that demonstrate the advantages of the present method?
2. The figures lack clarity and are difficult to understand. It is important to highlight the main contributions of the present modules.
3. The Semantic Integration Module appears to be resource-intensive, and it would be helpful to present the model parameters compared to previous methods.
4. The use of off-the-shelf designs in all the modules limits the novelty of the approach.

**Limitations:**

The limitations are only briefly mentioned in the current manuscript.

---

> ### Author Rebuttal · Authors · 2023-08-08
>
> We sincerely thank the Reviewer ihRG for his/her review time and valuable comments. To the concerns expressed in Weaknesses and Questions , our response is as follows:
>
> *Weakness-1*: Thanks for your suggestion. The main contribution and motivation of this paper is to understand video content from a global perspective and achieve video-level cross-modal alignment, which helps to handle descriptions about temporal variations as well as improve performance. Therefore, the Semantic Integration Module (SIM), especially the video-level object cluster, is the core component. It's worth noting that the Video Object Cluster (VOC) is a part of SIM, which may be the cause of your confusion. We have updated the framework figure to more clearly illustrate our contribution (shown in the rebuttal pdf). Besides, our SOC is a simple but effective framework. Specifically, the model are composed of feature encoding (encoder), multi-modal interaction (MMF), intra-frame content aggregation (SIM), inter-frame temporal modeling (SIM), and several output heads. Extensive comparisons of parameter, FLOPs, FPS, and performance show that our method outperforms previous state-of-the-art methods. Finally, the modules in Figure 1 have been described in the Section 3.3, *i.e.*, Frame-Level Content Aggregation and Video-Level Object Cluster. We also emphasize that in the updated framework figure.
>
> *Weakness-2*: Thanks for your great advice. Table 3 and Table 4 in the manuscript show the ablation about the VOC (temporal modeling part of SIM) and experiments on the MMF. Here we experiment with the effectiveness of each module on the whole network. "W/o MMF" indicates replacing MMF with simple cross attention operation, *i.e.*, the approach used in previous models. We will update these results in the manuscript.
>
> | Model | $\mathcal{J} $&$ \mathcal{F}$ | $\mathcal{J} $ | $\mathcal{F} $ |
> | :---: | :---: | :---: | :---: |
> | Full Model | 59.2 | 57.8 | 60.5 |
> | W/o SIM | 51.6 | 50.4 | 52.8 |
> | W/o MMF | 58.7 | 57.3 | 60.0 |
> | W/o VL | 58.0 | 56.5 | 59.5 |
>
>
> *Weakness-3*: Great suggestion! We provide the visualization of the impact of each module on the segmentation results and corresponding descriptions in the rebuttal pdf. We will incorporate it into the manuscript.
>
> *Question-1*: Previous methods perform multi-modal interaction in each frame separately, which leads to difficulty in comprehending the text description of temporal variations and actions. With regard to the two examples illustrated in Figure 6, in the first case, since the girl is **sitting** at the beginning, the compared method can not find an object that matches the description of **"a person is jumping over the river"** in this frame. For the second case, there are two athletes in the video (possibly requiring zooming in to discern the distant athlete). Due to the limitations of multimodal understanding at the frame level, previous method is unable to understand the temporal notion of "from right to left". Consequently, it may only understand other concepts such as "tennis player is running" to the best of their ability, and thus resulting in segmentation errors. Furthermore, in the supplementary material and demo, we showcase various of segmentation results in complex video scenarios *e.g.*, similar appearance, occlusion, and large variations, which demonstrates the generalization of our method  in comprehending the textual description about temporal variations.
>
> *Question-2*: Thanks for the suggestion. We will revise the manuscript to make the figures and corresponding descriptions clearer. The main contribution and motivation of this paper is to understand video content from a global perspective and achieve video-level cross-modal alignment, which allows the model can handle text descriptions about temporal variations.
>
> *Question-3*: Thanks for the advice. We perform detailed comparisons ((including model parameters)) in the table below. It can be seen that our SOC has a smaller number of parameters than the previous SOTA method ReferFormer. While MTTR has the lowest FLOPs and the fewest parameters, the lack of elaborate multi-modal fusion and temporal interaction significantly degrades segmentation accuracy.
> In addition, our Semantic Integration Module is computation-friendly and efficient: the number of parameters and the inference time consumed by SIM are **9.8%** and **8.3%** of the whole model, respectively.
>
> | Model | Backbone | Parameters | $\mathcal{J} $&$ \mathcal{F}$ | FPS | FLOPs (a video with 43 frames)|
> |-----|-----|-----|-----|-----|-----|
> | MTTR | Video-Swin-T | 104.88M | 55.3 | 31.8 | 1018G |
> | ReferFormer | Video-Swin-T | 131.67M | 56.0 | 21.2 | 2994G |
> | SOC (Ours) | Video-Swin-T | 131.46M | 59.2 | 32.3 | 1879G |
>
> *Question-4*: The motivation of this work is to unify temporal modeling and cross-modal alignment to achieve video-level understanding. Some network layers such as transformer layer are indeed off-the-shelf. However, we believe that using these network layers to realize our motivation and design an efficient framework to fuse intra- and inter-frame information for video-level multi-modal understanding and improving RVOS task is technically contributory.

---

> > ### Comment · Reviewer_ihRG · 2023-08-17
> >
> > I'm pleased with the response. The authors offer clear explanations and include extensive experiments to showcase the effectiveness of the proposed designs. All my concerns have been adequately addressed in the response. I'm happy to upgrade the rating to weak accept.

---

### Official Review · Reviewer_NrNc · 2023-07-06

**Soundness:** 3 good
**Presentation:** 2 fair
**Contribution:** 3 good
**Rating:** 6
**Confidence:** 4

**Summary:**

This article introduces the Semantic-assisted Object Cluster model, which is used to solve the problem of visual language alignment between videos and texts. The model uses a semantic integration module and visual language contrastive learning to improve the understanding of video content and has achieved significant improvements in different scenes and text descriptions.

**Strengths:**

1．The Semantic-assisted Object Cluster (SOC) framework has been proposed, which effectively aligns video and text, realizes object aggregation and visual-language alignment, and improves the performance of the RefVOS model.
2．The Semantic Integration Module (SIM) has been designed to efficiently aggregate intra-frame and inter-frame information, understand video content from a global perspective, and help understand temporal changes and alignment of different modalities and granularities.
3．Visual-linguistic contrastive learning has been introduced to provide semantic supervision, guide the establishment of video-level multimodal joint space, and improve the performance of the RefVOS model. 4、It can effectively handle text descriptions that express temporal changes and significantly improve general scene processing.
4.On popular RVOS benchmarks such as Ref-YouTube-VOS, Ref-DAVIS, A2D-Sentences, and JHMDB-Sentences, the performance is significantly better than existing methods, and inference speed is faster.


**Weaknesses:**

1．The effectiveness of Visual-Linguistic Contrastive Learning module heavily depends on the quality of the text descriptions used for supervision. If the text descriptions are inaccurate or incomplete, the performance of this module could be limited.
2．The efficiency of SIM is not clearly explained, and it is unclear how well it would scale to larger and more complex datasets.
3. The caption of many figures (such as Fig.3) are too simple, and more details should be provided.
4.Ablation experiments were conducted only within each module, without verifying the effectiveness of each module on the whole network framework.


**Questions:**

1．For the Video-Level Object Cluster module, the design of the object clustering module does not fully consider the similarities and differences between instances, which may affect the clustering effect. The paper should provide some considerations on this issue.

2．For the Visual-Linguistic Contrastive Learning module, the issue of potentially non-target responses in video-level embeddings due to the simple use of text priors should not be ignored.

3．The model uses Video-Swin-B and Video-Swin-T as backbones, so it is unfair to compare it with other models that use resnet as backbone.


**Limitations:**

1．The model may rely on the accuracy of semantic information, as it uses the Semantic Integration Module (SIM) to aggregate information in the video.

2．The model may not be able to handle some complex video scenes because it only performs inference based on known semantic information.

---

> ### Author Rebuttal · Authors · 2023-08-08
>
> We sincerely thank the Reviewer NrNc for his/her comments and the appreciation of our work. To the concerns expressed in Weaknesses and Questions, our response is as follows:
>
> *Weakness-1:* Good question! Inaccurate or incomplete descriptions do affect the effectiveness of Visual-Linguistic Contrastive Learning. To alleviate this problem, we adopt the V2L stream in MMF that updates the textual embedding with video content. Besides, such ambiguous descriptions would degrade the performance of all RVOS methods. In fact, the textual description is artificially controllable. Users can easily provide clear descriptions in most cases. Finally, our approach surpasses previous methods even without Visual-Linguistic Contrastive Learning module, thanks to our video-centric paradigm design.
>
> *Weakness-2:* Thanks for your suggestion. We further investigate the overhead of each module and present the results in the table below. Results show that SIM accounts for only 8.3% of the total inference time.
>
> | Modules | Backbone | MMF | Frame-Level Aggregation | Video-Level Clustering | Head | Total |
> | --- | :---: | :---: | :---: | :---: | :---: | :---: |
> | Time per frame (ms) | 10.4 | 0.41 | 2.48 | 0.098 | 0.46 | 30.9 |
>
>
> Besides, because the VOC module in SIM is central to video-level multimodal understanding, the ablation we perform in the manuscript is for the VOC module. Here we perform detailed experiments on the effectiveness of SIM and the results are shown below.
>
> | Model | No SIM | Only Frame-Level Aggregation | Only Video-Level Clustering | Entire SIM |
> | --- | :---: | :---: | :---: | :---: |
> | $\mathcal{J} $&$ \mathcal{F}$ | 51.6 | 57.1 | 54.7 | 59.2 |
>
> To the best of our knowledge, there are no larger scale annotated datasets available in this field. We have evaluated our model on all popular RVOS benchmarks and the performance is significantly better than existing methods. In addition, we also agree that large-scale RVOS datasets for complex scenarios are meaningful.
>
> *Weakness-3:* Thanks for your valuable comments! Due to page limitations in the review version (one page less than the final version), some captions may be too concise. We will revise the paper based on your suggestions to provide more details and make the caption clearer.
>
> *Weakness-4:* Thanks for your advice. We experiment with the effectiveness of each module on the whole network and the results are shown in the table below. "W/o MMF" indicates replacing MMF with simple cross attention operation, *i.e.*, the approach used in previous models. We will update these results in the manuscript.
>
>
> | Model | $\mathcal{J} $&$ \mathcal{F}$ | $\mathcal{J} $ | $\mathcal{F} $ |
> | :---: | :---: | :---: | :---: |
> | Full Model | 59.2 | 57.8 | 60.5 |
> | W/o SIM | 51.6 | 50.4 | 52.8 |
> | W/o MMF | 58.7 | 57.3 | 60.0 |
> | W/o VL | 58.0 | 56.5 | 59.5 |
>
>
> *Questions-1:* On the one hand, frame-level content aggregation highlights the information about the target object in each frame, enabling most queries to perceive the referred object (please see query visualization in the supplementary materials). This can effectively suppress noise caused by other instances during the fusion process. On the other hand, the transformer architecture has powerful feature recognition and extraction capabilities, which are also helpful in generating desirable video-level object embeddings. We agree that explicitly considering similarities and differences between instances may yield better video-level object representations, which is an interesting research direction. It will be our future work.
>
> *Questions-2:* Thanks for the question. The design of V2L stream in MMF is to relieve the potential ambiguity of unconstrained descriptions by incorporating visual content into linguistic embeddings. Besides, the Visual-Linguistic Contrastive Loss is designed to relieve the potentially non-target responses in video-level embeddings by discerning the optimal query for emphasis and suppressing the response of other queries. As for the potential effect of imperfect textual descriptions on Visual-Linguistic Contrastive Loss, please refer to the response to *Weakness-1*.
>
> *Questions-3:* Here we provide the comparisons of our method with others using ResNet-50  as backbone. Results show that our SOC also outperforms previous methods.
>
>
> | Method | $\mathcal{J} $&$ \mathcal{F}$ | $\mathcal{J} $ | $\mathcal{F} $ |
> | :---: | :---: | :---: | :---: |
> | URVOS | 47.2 | 45.3 | 49.2 |
> | LBDT-4 | 49.4 | 48.2 | 50.6 |
> | ReferFormer | 52.1 | 51.4 | 52.8 |
> | SOC (Ours) | **54.8** | **53.7** | **55.9** |
>
> *Limitations:* Thanks for your acknowledgement of our work. Sorry that we are not quite sure whether the semantic information here refers to the text description or the video content. If text descriptions, please see the response to *Weakness-1*. If video content or object queries, we believe the limitations mentioned here may not be significant. On the one hand, we propose frame-level content aggregation to efficiently extract information about the target object and suppress the background region (visualization of query can be seen in the supplementary materials). On the other hand, the superiority of the learnable query representation paradigm has been proven in many tasks. Related methods such as Deformable DETR and Mask2Former have also shown the generalization ability to different scenarios. We wonder if the above response addresses your questions or concerns. If you still have any questions about our response, we would be pleased to discuss them with you.

---

> > ### Comment · Reviewer_NrNc · 2023-08-20
> >
> > Thanks for the authors's responses. I would like to keep my initial rating '6: Weak accept'.

---

### Official Review · Reviewer_8Kjg · 2023-07-12

**Soundness:** 2 fair
**Presentation:** 3 good
**Contribution:** 2 fair
**Rating:** 6
**Confidence:** 4

**Summary:**

This paper mainly studies referring video object segmentation (RVOS) which is a fundermental problem in vision-and-language applications. Compared with previous works which pay little attention to temporally modeling relationships between different frames in the entire video clip, the authors proposed a framework called SOC for RVOS which conducts temporal modeling and cross-modal alignment in a unified framework. Specifically, SOC contains several modules including textual and vision encoding and multi-modal fusion, semantic integration module and vision-language contrastive learning. The experimental results show SOC outperforms existing SOTA method, e.g. ReferFormer published in CVPR 2022.

**Strengths:**

(1) Overall, the motivation for poposing the temporal modeling intra-frame and inter-frame information for RVOS is quite clear and straightforward. Besides, the paper is well-writen and the idear as well as each module is easy-to-follow.

(2) The most valuable contribution is the video-level object cluster module in the semantic integration module (SIM).  Instead of using traditional clustering method such as k-means, the authors proposed an object grouping decoder architecure to perform the clustering process. Specifically, N_v video-level object queries are introduced to learn the overall object centroids in the entire video. After repeating such operation T times, the semantic representation enhancement is achieved by enriching the video-level object information. Figure 1 in the supplementary material demonstrates that most of these learnable queries focus on the region-of-interests (ROIs)  of he referred objects.

(3) The experimental results are promising. The proposed SOC framework outperforms other existing SOTA works under different settings. The provided video demo shows the referred objects have been segmented perfectly.

**Weaknesses:**

(1) According to my understanding of this work, the main contributions is the video-level object cluster module. Hence, the albation studies or comparison between different methods should focus on such module. Specifically, is there any other clustering-based method that could obtain video-level object embeddings? Why is the proposed object grouping decoder-style model is the best solution among these optional methods? Besides such high level comparisons, some details (or parameters) of abaltion studies should be further discussed. For example, the number of object queries and number of temporal frames are tuned independently in Table 5 and 6. However, jointly tuning these two parameters should be investigated.  Another example is that the final representation is the element-wis sum of frame-level and video-level semantic embeddings. Are there any other better learning-based fusion implementations?

(2) Generally, the proposed framework follows a two-stream fashion which includes a textual encoder and a vision encoder. Note that the vision encoder could be a temporal convolutional network (3D CNN) or video transformer such as in this work. No matter which implementation, it undergoes the temporal processing. The question is: could the video-level object clustering module be embedded inside such temporal encoding procedure? I think the whole inferece speed could be further decreased if it is possible.

(3) I think the main contribution and the motivation is clear.  Some trivial parts such as the visual-linguistic contrastive learning could be removed in order to make the key points stand out, which will not reduce the whole work's contribution or value.

**Questions:**


(1) Why the SIM module is not designed inside the video encoder? Please refer to the second part in the above weaknesses.

(2) It is reported that the SOC runs at 32.3 FPS on a single 3090 GPU. I think the overhead of each module should be further decomposed, e.g. the encoding backbone, multi-modal fusion module, and the SIM.

(3) As illustrated in Table 4, L2V outperforms V2L significantly. Deeper insights are expected to be uncovered. Although using both of them achieves the best performance, some missing details should be claimed. For example, what is each branch's weight?

(4) Could the proposed SOC be extended to solve multiple referred objects given a textual query?

**Limitations:**

The authors have mentioned that malicious use of such model could lead to negative social impacts such as unauthorized video surveillance. However, I believe the positive impacts of these vision-and-language applications would improve the embodied AI research and generally outweigh the risks.

---

> ### Author Rebuttal · Authors · 2023-08-08
>
>
>
> We sincerely thank the Reviewer 8Kjg for his/her comments and the appreciation of our work. To the concerns expressed in Weaknesses and Questions, our response is as follows:
>
> *Weakness-1:* (1) Thanks for your valuable question. Table 7 in the manuscript shows the ablation of the structure of the current video-level object cluster (VOC) module.  Here we experiment with several intuitive methods to obtain video-level object embeddings. Specifically, "Average" means global pooling of all frames-level queries at the corresponding positions. "1D-Conv" denotes that we use 1D convolution (kernel size is 5) to process the frame-level query and then pooling them. It can be seen that K-Means may lead to information confusion.
>
> | VOC Method | Average | K-Means | 1D-Conv | Ours |
> | --- | :---: | :---: | :---: | :---: |
> | $\mathcal{J} $&$ \mathcal{F}$| 57.5 | 33.8 | 55.1 | 59.2 |
>
> (2) We believe that the superiority of the current VOC module derives from two aspects: i) Powerful representation learning ability of the transformer structure. Compared to other clustering methods, the transformer architecture has a better perception as well as selection of global information, which helps to associate object features across frames. ii) Sufficient inter-frame interactions. Although other methods can also yield video-level representations, they lack the perception of temporal variations and the alignment of inter-frame information. As a contrast, our VOC performs adequate inter-frame interaction while clustering.
>
> (3) Thanks!  Here we involve more experiments on jointly tuning the object queries and temporal frames.
>
> | Query Num. | Frame Num. | $\mathcal{J} $&$ \mathcal{F}$ |
> | --- | --- | --- |
> | 10 | 3 | 58.1 |
> | 15 | 5 | 57.9 |
> | 15 | 8 | 58.1 |
> | 20 | 8 | 59.2 |
> | 25 | 3 | 57.8 |
> | 25 | 10 | 58.4 |
>
> (4) We experiment with two learning-based fusion methods for comparison. Self-Attn: we concatenate the frame-level and video-level queries along the sequence length dimension to obtain a new sequence, then perform self-attention operation to this sequence, and extract the corresponding frame-level query according to the index. Cross-Attn: we take frame-level object embeddings as the query, and the video-level embeddings as the key and value. Results are shown as below. It can be seen that the performance of the learnable fusion methods is about the same as element-wise sum.
>
> | Fusion Method | Add | Self-Attn | Cross-Attn |
> | --- | :---: | :---: | :---: |
> | $\mathcal{J} $&$ \mathcal{F}$| 59.2 | 59.1 | 58.9 |
>
> *Weakness-2:* Interesting question! This is a valuable research direction. Limited by rebuttal time and computation resources, we are unable to provide some experiment results at this time. Nevertheless, we have thoroughly considered the matter. We believe that there exists three potential challenges to be addressed for this idea. i) The lack of multi-modal interaction within the visual backbone may lead to noise when aggregating different frame embeddings. Specifically, without highlighting the target objects (*i.e.*, the MMF module and frame-level content aggregation), video-level information fusion  may lead to feature confusion due to the diversity of video scenes. ii) Computation cost. Since inter-frame interactions are performed by compact representations, *i.e.*, object query, our VOC is computation-friendly and efficient. Video-level clustering inside the encoder is performed on feature maps, which may lead to huge computations. iii) Modifying the structure of backbone may disrupt the pretrained parameter relationship and potentially impair the backbone's ability. We will further explore this in our future work.
>
> *Weakness-3:* Thanks for your advice and appreciation of our work! We will integrate the visual-linguistic contrastive loss into the introduction of total loss in Section 3.5. Additionally, we will strive to provide more comprehensive explanations and emphasize our primary contribution.
>
> *Question-1:* Please see the response to *Weakness-2*.
>
> *Question-2:* Thanks for your great suggestion. We decompose each module of our method and report the overhead in the table below. It can be seen that our VOC module is efficient. This also shows the rationality and advantages of the design of our VOC module.
>
> | Modules | Backbone | MMF | Frame-Level Aggregation | Video-Level Clustering | Head | Total |
> | --- | :---: | :---: | :---: | :---: | :---: | :---: |
> | Time per frame (ms) | 10.4 | 0.41 | 2.48 | 0.098 | 0.46 | 30.9 |
>
> *Question-3:* (1) Deeper explanation: L2V stream highlights the corresponding regions of the referred object in each frame and mitigates the impact of background noise. It serves as an important pre-fusion procedure to enable the model to focus more on the referred object in each frame so that the following SIM module can extract the referred object embeddings and aggregate intra-frame and inter-frame information. Without the L2V stream, the model loses the perception of the referred object and can hardly output desired segmentation results. In terms of the V2L stream, it updates the textual embedding with image content, which helps to relieve the potential ambiguity of unconstrained descriptions. Without the V2L stream, the model is still able to perceive the referred object. Therefore, L2V outperforms V2L significantly. (2) Details: This question may leave us a little confused. Figure 3 in the manuscript shows that the V2L and L2V have their own function and operate separately without any balancing weights. The attention weights in the figure refers to the attention map computed by query and key. If you still have any questions, we would be glad to further discuss with you.
>
> *Question-4:* We think it's OK. Because multiple objects can be perceived by multiple object queries. The head and cross-modal interaction module may need to be modified accordingly. The main limitation may be that the current dataset and settings of RVOS task only involve single target object.

---

> > ### Comment · Reviewer_8Kjg · 2023-08-11
> >
> > Thanks for the authors' rebuttal. I think my concerns proposed in the review of submission have almost been resolved.

---

> > > ### Author Response · Authors · 2023-08-12
> > > **Official Comment by Authors**
> > >
> > > Dear Reviewer 8Kjg,
> > >
> > > Thanks for your response. We are glad that our rebuttal is able to address your concerns. We sincerely appreciate your review time and the valuable comments. We will  attach great importance to revising the paper according to your suggestions.
> > >
> > > Best regards,
> > >
> > > Paper 4195 Authors

---

> > > > ### Comment · Reviewer_8Kjg · 2023-08-18
> > > >
> > > > I have already upgraded the reating. Good luck.

---

### Official Review · Reviewer_iFku · 2023-07-24

**Soundness:** 3 good
**Presentation:** 3 good
**Contribution:** 3 good
**Rating:** 4
**Confidence:** 2

**Summary:**

In this paper, the author presents a framework called SOC, and a visual-linguistic contrastive loss for RVOS to unify temporal modeling and cross-modal alignment.
The performance is good according to the leaderboard (https://paperswithcode.com/sota/referring-video-object-segmentation-on-refer)
The extra module seems to have many parameters, and somehow incremental.

**Strengths:**

In this paper, the author presents a framework called SOC, and a visual-linguistic contrastive loss for RVOS to unify temporal modeling and cross-modal alignment.
The performance is good.

**Weaknesses:**

1. Parameters
The extra module seems to have many parameters, and somehow incremental.
- Unfair Comparison. It is unfair to compare with other methods.
- Design of MMF is not new. Many multi-modality fusion has used similar structure such as [a]. The different  part is that authors adopt transformer structure.
[a] Comprehensive Linguistic-Visual Composition Network for Image Retrieval

2. Motivation in Figure 1 is not clear to me.
I think it can be easily solved by  adopting the video-based backbone such as TSM or I3DNet,

3. The visual-linguistic contrastive loss
I do not see significant difference from the basic contrastive loss in Clip or other works. The input is changing from image to video features. Could you further explain it?



**Questions:**

Please see Weakness. I am not working in this field. If the author could address my concerns, I will upgrade my rate.

**Limitations:**

Please see Weakness. I am not working in this field. If the author could address my concerns, I will upgrade my rate.

---

> ### Author Rebuttal · Authors · 2023-08-07
>
> We sincerely thank the Reviewer iFku for his/her comments and the appreciation of our work. To the concerns expressed by Reviewer iFku in Weaknesses and Questions , our response is as follows:
>
> *Weakness-1:* Thanks for reminding us that we should make a clear demonstration of model parameters. In our supplementary materials, we compared our method with previous state-of-the-art RVOS approaches (MTTR, ReferFormer) in terms of performance, FPS, and FLOPs. Here we add parameter comparisons and list comprehensive data in the table below. Our method achieves superior performance with faster inference speed and fewer parameters compared to the previous SOTA method Referformer. While MTTR has the lowest FLOPs and the fewest parameters, the lack of elaborate multi-modal fusion and temporal interaction significantly degrades segmentation accuracy.
>
> | Model | Backbone | Parameters | $\mathcal{J} $&$ \mathcal{F}$ | FPS | FLOPs (a video with 43 frames)|
> |-----|-----|-----|-----|-----|-----|
> | MTTR | Video-Swin-T | 104.88M | 55.3 | 31.8 | 1018G |
> | ReferFormer | Video-Swin-T | 131.67M | 56.0 | 21.2 | 2994G |
> | SOC (Ours) | Video-Swin-T | 131.46M | 59.2 | 32.3 | 1879G |
>
> - Unfair Comparison: Thanks for the question. We would like to confirm whether the question about potential unfair comparisons stems from your concern about the parameters? If so, the above table demonstrates that our method achieves higher segmentation accuracy with fewer parameters compared to the previous SOTA method ReferFormer. Besides, all comparisons in the manuscript are based on fair settings.
>
> - Design of MMF is not new: Thanks for the question. MMF is just a component of our framework. It is not our contribution, and it is not mentioned in the contribution note in the introduction section of the paper. Our main contribution lies in the proposed framework, which unifies temporal modeling and cross-modal alignment to achieve video-level visual-linguistic understanding. The core module of our framework is the Semantic Integration Module. By fusing intra-frame and inter-frame information, our approach achieves better performance and can handle descriptions of temporal variations. Regarding the MMF module specifically, it serves to activate textual referred objects in visual features and simultaneously update the textual embeddings by incorporating visual content.
>
> We wonder if the above response addresses your questions about the model parameters and the incremental modules. If you still have any questions about our response, we would be pleased to discuss them with you.
>
> *Weakness-2:* Good question. TSM or I3D can indeed provide temporal information. However, segmentation tasks need spatially fine-grained features and emphasize pixel-level understanding. I3D or TSM can only provide coarse-grained features (frame-level or video-level). Using I3D directly as the backbone would significantly degrade segmentation accuracy. That's why I3D is not used in video segmentation. In addition,  a viable option that can take advantage of the ability of I3D to provide video-level information may be to employ two backbones at the same time, *i.e.*, a spatial backbone and a temporal backbone (I3D). This approach, undoubtedly, introduces an excessive number of parameters and computational costs. In contrast, our method realizes the fusion of intra-frame fine-grained information and inter-frame temporal relations efficiently.
>
> *Weakness-3:* Thanks for the question. We believe that the form of the visual-linguistic loss is not significant, but rather its motivation and the supervision object. Specifically, this loss is applied to further build the multi-modal joint space at the video level. Unlike CLIP that performs  contrast learning  between different images and captions inside the training batch, what we want to establish is the contrast between different object queries and the same text descriptions. Furthermore, we would like to emphasize that the main contribution of this paper is to unify temporal modeling and cross-modal alignment to achieve video-level visual-linguistic understanding and excellent segmentation performance. In other words, the design of the pipeline, especially the Semantic Integration Module, is the core of this paper.

---

> > ### Comment · Reviewer_iFku · 2023-08-14
> > **More explaination on Semantic Integration Module**
> >
> > Thank you.
> >
> > 1. Could you provide more details?
> > - How about the text backbone? Are they the same for the three works?
> > - Why the proposed SIM is good for fine-grained details? How do you define the fine-grained and coarse-grained for VOS?
> >
> > 2. Yes. Could you try TSM? I think TSM can do good efficiency in term of Figure 1. Why TSM is coarse-grained?
> >
> > 3.   Could you explain more on "the contrast between different object queries and the same text descriptions" ?
> > I think it is still basic contrastive learning idea.

---

> > > ### Author Response · Authors · 2023-08-16
> > > **Response to Reviewer iFku (Part 1/2)**
> > >
> > > Thanks for your feedback. The following is our further explanation:
> > >
> > > **Response to 1.1**: The text backbone is RoBERTa. Yes, the text encoder is the same for the three works.
> > >
> > > **Response to 1.2**: Coarse-grained and fine-grained mentioned in our rebuttal refer to the **spatial** understanding level of the visual content. Specifically, coarse-grained means that the model understands the general content of the image or video, while fine-grained means that the model understands the image at the pixel level. Here is an example:  given a skateboarding video, the coarse-grained understanding method can only understand that the content of this video is a person skateboarding. While fine-grained understanding requires distinguishing different instances as well as accurately recognizing the shape, position, edges, etc. of objects in each frame.
> > >
> > > As can be seen, spatially fine-grained understanding is more demanding, which leads to the need for more discriminative pixel-level or object-level features. This is the strength of our SIM: the ability to perform temporal modeling while guaranteeing the discriminative representations (spatial modeling ability). We will explain this in more detail in the answer to question 2 below.
> > >
> > > **Response to 2**: Thanks for your question. Because TSM [1] is a plug-and-play module and the paper provides TSM-ResNet50 backbone weights, we have tried TSM in two ways: i) take the TSM-ResNet50 as backbone (meanwhile removing our temporal modeling part) and ii) replace our video object clustering  module (temporal modeling part) by TSM.
> > >
> > > *i) Take the TSM-ResNet50 as backbone (our temporal modeling part is removed, our frame-level aggregation part is preserved):*
> > >
> > > | Method | $\mathcal{J}$&$\mathcal{F}$|
> > > |---------|:--------:|
> > > | TSM-ResNet50 Backbone (ImageNet Pretrained)  | 50.3 |
> > > | TSM-ResNet50 Backbone (Kinetics-400 Pretrained) | 51.2 |
> > > | Ours (ResNet50, ImageNet Pretrained))  | 54.8 |
> > >
> > >
> > > The ImageNet pretrained TSM-ResNet50 indicates that the ResNet layers are loaded with pretrained weights on ImageNet. While the TSM layers embedded in ResNet are trained from scratch. The Kinetics-400 pretrained TSM-ResNet50 denotes that we load the full backbone weights provided in the TSM paper. It can be seen that even after pre-training on the video dataset, simply replacing the backbone network is still difficult to achieve excellent performance. We believe this is for two main reasons:
> > > - (a) The operation of shifting channels used in TSM will hurt the spatial modeling ability of models (the TSM paper also recognizes this). Specifically, TSM performs temporal modeling at the cost of introducing noise and leading to feature confusion. Simply shifting channels along temporal axis breaks the coupling of features and leads to spatial information confusion, which greatly impairs the discriminability of features and the precise identification of instances. To alleviate this problem, the TSM paper shifts only a small number of channels and insert TSM through residual connections.  While this helps to some extent with coarse-grained understanding, it is still suboptimal for fine-grained understanding tasks because the feature obfuscation problem remains unresolved. As mentioned above, spatially fine-grained understanding is more demanding, which leads to the need for more discriminative pixel-level or object-level features. In contrast, our SIM integrates intra- and inter-frame information well.
> > > - (b) The TSM backbone aggregates different frame features before multi-modal interaction, which may lead to feature confusion and noise due to the diversity of video scenes. Specifically, without highlighting the target objects, inter-frame interaction is disorganized and is affected by background noise (non-target regions).
> > >
> > > *ii) Replace our temporal modeling part by TSM (our frame-level aggregation part is preserved):*
> > >
> > > | Method | $\mathcal{J}$&$\mathcal{F}$|
> > > |---------|:--------:|
> > > | Replace our temporal modeling part by TSM   | 57.5 |
> > > | SOC (Ours) | 59.2 |
> > >
> > > It can be seen that TSM is inferior to our video clustering module. We believe this is also for two main reasons. The first is the same as (a) above, *i.e.*, The operation of shifting channels used in TSM will hurt the spatial modeling ability of models. The second is that TSM can only perform local temporal modeling, while our approach can perform global (video-level) information interaction. Experiment results in our rebuttal to Reviewer d9RC's last question also prove that the global interaction is better.

---

> > > > ### Author Response · Authors · 2023-08-16
> > > > **Response to Reviewer iFku (Part 2/2)**
> > > >
> > > > In conclusion, TSM can indeed perform temporal modeling efficiently. However, the information confusion and noise caused by shifting channel operations can also harm the video segmentation task and degrade the model performance. Besides, The statement in the original rebuttal may be a bit confusing.TSM is not coarse-grained. But as mentioned above, TSM is inferior to fine-grained tasks such as segmentation due to the impairment of spatial modeling ability. TSM may be more suited to coarse-grained tasks such as video classification. In comparison, our VOC (temporal modeling part in SIM) aligns more seamlessly with the RVOS task.
> > > >
> > > > Now we can elaborate on why our SIM is good for fine-grained details. Firstly, our SIM is composed of frame-level aggregation and video-level clustering. Such design allows the SIM to integrate intra- and inter-frame information well. Besides, the frame-level aggregation part extracts object information into compact representations, *i.e.*, object query embeddings, which greatly reduces the effect of background noise on the temporal modeling.  Secondly, compared to other temporal modeling methods, the superiority of our VOC module (temporal modeling part of SIM) derives from two aspects: i) Powerful representation learning ability of the transformer structure. The transformer architecture has a better perception as well as selection of global information, which helps to associate object features across frames. ii) Reasonable video-level interactions. Learnable temporal modeling approaches have greater potential than hard association strategies such as TSM. Meanwhile, since our VOC is performed on compact representations, it does not bring heavy computation cost.
> > > >
> > > > **Response to 3**: Our contrastive loss is not simply changing the input of CLIP loss from image to video features. It is worth noting that our contrastive loss is **intra-video not inter-video**. Both positive and negative samples in our contrastive loss are **from the same video**. But CLIP performs contrastive learning between different images (videos) within the training batch. Specifically, the input of our contrastive loss is a group of video object queries that contain information about the target object in the same video. In contrast to CLIP which has a clear definition of positive and negative samples, since all these object queries come from the same video, we need to use a matching algorithm to find out the most appropriate query embedding to be positive sample. Such dynamic positive sample strategy and intra-video contrastive learning enhance the discrimination of object query features and reduce noise interference, which is helpful for recognizing and segmenting target objects.
> > > >
> > > > Thanks again for your feedback. We hope that our response can address your questions, and if you still have any concerns, we would be pleased to discuss them further with you.
> > > >
> > > > [1] TSM: Temporal Shift Module for Efficient Video Understanding

---

> > > ### Author Response · Authors · 2023-08-21
> > > **Further discussion with Reviewer iFku**
> > >
> > > Dear reviewer iFku:
> > >
> > > We thank you for the precious review time and comments. We have provided corresponding responses and results, which we believe have covered your concerns. We hope to further discuss with you whether or not your concerns have been addressed. The discussion period ending date is August 21. Please let us know if you have any unsolved or other concerns.
> > >
> > > Thanks,
> > >
> > > Paper 4195 Authors.

---

### Author Rebuttal · Authors · 2023-08-09

Dear Reviewers and ACs:

We sincerely appreciate all the reviewers. They give positive and high-quality comments on our paper with a lot of constructive feedback. We are glad to be recognized that our motivation is clear and straightforward (R2, R3, R6), the paper is well-written and easy-to-follow (R2, R6), the proposed modules helps understand temporal changes and alignment of different modalities and granularities (R3, R5), the performance is significantly better than existing methods with faster inference speed (R1, R3, R4, R5), and the experimental results are promising (R2).

We respond to each reviewer in the separate rebuttal part. If our response does not fully address the reviewer's questions, we are pleased to discuss further with the reviewers.

Besides, we will revise our manuscript according to all the reviewers' comments. Thank you all for the valuable suggestions.

Thanks,

Paper 4195 Authors.

---

### Decision · Program_Chairs · 2023-09-21

**Decision:**

Accept (poster)

**Comment:**

The submission focuses on the referring video object segmentation (RVOS) task, whose goal is to segment an target object per video frame given a text description. The main motivation of the submission is to construct video-level object-embedding which captures temporal information, and the main contribution claimed by the authors is a unified framework with temporal modeling, spatial understanding, and cross-modal alignment for the RVOS task.

Overall, the reviewers found the submission to be well written, with clearly presented and motivated solutions, and offering "remarkable" performance on various benchmarks. However, they had concerns on the experiment design (e.g. if the number of parameters comparable with baselines), the lack of ablation on the video object clustering modules, and that further clarification is needed when presenting the primary contributions. The authors provided detailed responses to the reviewers, and four out of the five reviewers found their concerns to be effectively addressed, and recommended acceptance. Reviewer iFku had remaining concerns on the details of SIM and comparisons with TSM video encoder, and the AC believes that the authors provided sufficient explanations and supporting experiments to address the remaining concerns.

The AC thus recommends acceptance of the submission, and encourages the authors to incorporate the additional experiments during the rebuttal, and the reviewers' feedback into the final version.